# Hidden prevalence of deletion-inversion bi-alleles in CRISPR-mediated deletions of tandemly arrayed genes in plants

Jiuer Liu [1,2], Feng-Zhu Wang[1,2], Chong Li [1], Yujia Li[1] & Jian-Feng Li [1] ✉

Tandemly arrayed genes (TAGs) with functional redundancy and chromosomal linkage constitute 14 ~ 35% in sequenced plant genomes. The multiplex CRISPR system is the tool of choice for creating targeted TAG deletions. Here, we show that up to ~80% of CRISPR-mediated TAG knockout alleles in *Arabidopsis* and rice are deletion-inversion (delinver) bi-alleles, which are easily misidentified as homozygous deletion alleles by routine PCR-based genotyping. This can lead to misinterpretation of experimental data and production of progenies with genetic heterogeneity in an unnoticed manner. In ~2,650 transgenic events, delinver mutation frequencies are predominantly correlated with deletion frequencies but unrelated to chromosomal locations or deletion sizes. Delinver mutations also occur frequently at genomic non-TAG loci during multiplexed CRISPR editing. Our work raises the alarm about delinver mutations as common unwanted products of targeted TAG deletions in plants and helps prevent false interpretation of plant TAG functions due to this hidden genotype issue.

Tandemly arrayed genes (TAGs), which are paralogous genes with physical linkage to one another on a chromosome, are widely distributed in sequenced plant genomes[1]. The proportion of TAGs in *Arabidopsis thaliana*, rice, maize, and poplar genomes corresponds to 17%, 14%, 35%, and 16%, respectively[2–5]. Plant TAGs are overrepresented by genes involved in biotic and abiotic stress responses[6], as exemplified by those encoding disease resistance (R) proteins[7] and receptor-like kinases[8]. Strikingly, approximately 73% of *Arabidopsis R* genes and 76% of rice *R* genes belong to TAGs[9,10]. In functional genomics of TAGs, genetic redundancy and close chromosomal linkage have together posed demanding challenges. On one hand, the genetic redundancy of TAGs requires the knockout of multiple genes to obtain discernable mutant phenotypes carrying functional indications. On the other hand, the close linkage of those genes has hindered efficient generation of high-order knockout mutants by crossing single mutants of individual genes.

An effective strategy for interrogating TAGs is to generate large chromosomal deletions spanning the TAG locus of interest. A pioneer effort by Li and co-workers has successfully isolated a deletion allele of TAGs encoding multiple TGA transcription factors in *Arabidopsis* via the fast neutron-induced chromosomal deletion and PCR-based screening[11]. However, the random nature of chromosomal lesions induced by ionizing radiation makes it difficult to create TAG deletion mutants at will. By contrast, the site-specific nucleases, including zinc finger nucleases (ZFNs), transcription activator-like effector nucleases (TALENs), and the clustered regularly interspaced short palindromic repeats (CRISPR)/CRISPR-associated (Cas) system, can facilitate the generation of targeted TAG deletions in plants. Both ZFNs and TALENs are hybrid nucleases of a programmable DNA-binding domain and a *Fok*I nuclease domain[12,13]. When a pair of ZFNs or TALENs bind to two proximal DNA sequences on opposite strands, the two approaching *Fok*I nucleases can dimerize to cleave the intervening DNA to generate a double-strand break (DSB). The CRISPR system utilizes a guide RNA (gRNA) to base pair with the target DNA, enabling the gRNA-associated Cas nuclease to create a DSB at the target site[14]. When two DSBs are formed concurrently on

[1]State Key Laboratory of Biocontrol, Guangdong Provincial Key Laboratory of Plant Resources, MOE Key Laboratory of Gene Function and Regulation, School of Life Sciences, Sun Yat-sen University, Guangzhou 510275, China. [2]These authors contributed equally: Jiuer Liu, Feng-Zhu Wang. ✉e-mail: lijfeng3@mail.sysu.edu.cn

the same chromosome, there will be a chance to excise the intervening DNA, leading to targeted genomic deletions[15]. As a brilliant example, Qi and colleagues employed a pair of ZFNs to cleave two highly homologous DNA sequences at the TAG locus of interest to achieve deletions of TAGs spanning up to 55 kilobases (kb) in *Arabidopsis*[16]. Because the gRNAs in the CRISPR system can be modified more easily than the proteinaceous DNA-binding modules in ZFNs or TALENs to gain new DNA targeting specificities and because several gRNAs can collaborate with the same Cas nuclease to cleave multiple target sites, the CRISPR system offers unparalleled simplicity and multiplexability. By using a pair of gRNAs to target the two outermost genes at the TAG locus of interest, the CRISPR system has been proven highly effective for inducing TAG deletions in plants[17–26].

The paired gRNA strategy has also been explored to induce intended chromosomal rearrangements in plants, including chromosomal inversions and translocations[27–31]. These successes open up new opportunities for chromosomal engineering in crop molecular breeding[32–34]. Meanwhile, a growing body of research has raised concerns about unwanted genomic rearrangements induced by CRISPR in mammalian cells[35–38]. Similarly, unexpected products generated by CRISPR have recently been noticed in plants[39,40].

In this study, we report that deletion-inversion bi-alleles can be generated at unexpectedly high frequencies during CRISPR-mediated TAG deletions in *Arabidopsis* and rice. As this type of mutations can be easily misidentified as homozygous TAG deletion alleles during standard PCR-based genotyping to mislead the follow-up functional studies, we propose a modified genotyping PCR scheme to help distinguish these unwanted products from bona fide TAG knockout alleles.

## Results

### Deletion-inversion bi-alleles occur unexpectedly in CRISPR-mediated deletion of multiple *AtPROPEP*s

In *Arabidopsis*, the PROPEP phytocytokines are encoded by a gene family of eight members[41], in which *AtPROPEP8/7/4/5* and *AtPROPEP2/1/3* are two TAG loci on chromosome 5, while *AtPROPEP6* resides on chromosome 2 (Fig. 1a). To interrogate their functions in plant immunity and circumvent genetic redundancy, we employed the multiplex CRISPR system to generate *atpropep1-8* octuple mutant plants by co-expressing the *Streptococcus pyogenes Cas9* (*SpCas9*) with three pairs of gRNAs (Fig. 1b). The gRNA-Pep8 and gRNA-Pep5 targeted the TAGs of *AtPROPEP8/7/4/5*, the gRNA-Pep2 and gRNA-Pep3 were aimed at the TAGs of *AtPROPEP2/1/3*, while the gRNA-Pep6.1 and gRNA-Pep6.2 were used to knock out *AtPROPEP6*. We followed a standard two-tier PCR-based genotyping procedure to identify homozygous TAG deletion alleles of *AtPROPEP8/7/4/5* and *AtPROPEP2/1/3* in transgenic $T_1$ generation, in which short amplicons using two primers flanking the TAG locus (i.e., Fw1/Rev1) indicated the presence of TAG deletions, while negative amplicons using the same forward primer and a reverse primer annealing to the TAG locus (i.e., Fw1/Rev2) indicated the absence of intact TAG sequences (Supplementary Fig. 1a). Presumptive homozygous deletion alleles were further validated by Sanger sequencing of the PCR amplicons. As a result, we identified three *atpropep1-8* mutant lines, namely #2, #11, and #13. All three lines were supposed to carry the same homozygous 5.5- and 4.8-kb genomic deletions spanning the *AtPROPEP8/7/4/5* and *AtPROPEP2/1/3* loci, respectively (Supplementary Fig. 1b). In the lines #2 and #11, the mutated *AtPROPEP6* contained 1-bp insertion or deletion at the gRNA-Pep6.2 target site, while the line #13 harbored biallelic mutations, namely 1-bp insertion on one chromosome and 2-bp deletion on the other homologous chromosome, at the gRNA-Pep6.1 target site (Supplementary Fig. 1c).

The three *atpropep1-8* $T_1$ lines were phenotypically indistinguishable from wild-type (WT) plants (Supplementary Fig. 1d).

However, when conducting RNA sequencing (RNA-seq) analysis using their $T_2$ generation, we surprisingly noted that *AtPROPEP1*, which should have been deleted along with the TAGs of *PROPEP2/1/3* (Supplementary Fig. 1a, b), was still transcribed in the $T_2$ line #2-4 but not in #13-1 (Fig. 1c). By contrast, *AtPROPEP4* and *AtPROPEP7*, which should have been deleted like *AtPROPEP1*, exhibited no detectable expression in both lines as expected (Fig. 1c). It has been reported that the expression of endogenous *AtPROPEP*s can be induced upon treatment of the bacterial elicitor flg22 to amplify the flg22-triggered immunity[42]. Consistently, a group of AtPEP1 (AtPROPEP1-derived mature phytocytokine) responsive genes were still induced by flg22 in *atpropep1-8* #2-4 to a comparable level as in WT plants, whereas the induction of those genes by flg22 was completely abolished in *atpropep1-8* #13-1 plants (Fig. 1d). These findings suggest that only *atpropep1-8* #13-1 but not #2-4 is a bona fide null allele for *AtPROPEP1*.

To explain the perplexing observations, we reasoned that the deleted *AtPROPEP2/1/3* fragment might have been re-inserted into the genome in the *atpropep1-8* line #2-4. In line with this speculation, we obtained PCR amplicons using primer pairs spanning the *AtPROPEP1* locus in *atpropep1-8* #2-4 as in WT plants (Supplementary Fig. 2a). To define the re-insertion site, we conducted *AtPROPEP1*-based thermal asymmetric interlaced PCR (TAIL-PCR) (Supplementary Fig. 2b, c) and Sanger sequencing. Interestingly, the sequencing results revealed that the deleted *AtPROPEP2/1/3* fragment was invertedly re-inserted between the gRNA-Pep2 and gRNA-Pep3 induced DSBs in *atpropep1-8* #2-4 (Fig. 1e and Supplementary Fig. 2d). To consolidate this finding, we carried out an additional PCR using two co-aligned primers (i.e., Fw1/Fw2 or Rev1/Rev2) to amplify the *AtPROPEP2/1/3* locus (Fig. 1e). These co-aligned primers would fail to produce any amplicons in case the chromosomal region was retained in the WT orientation, but would face each other to generate amplicons when an inverted re-insertion event occurred. Indeed, PCR amplicons with expected sizes were detected only in *atpropep1-8* #2-4 but not in #11-1 or #13-1 plants (Fig. 1f). Sanger sequencing of the amplicons further validated the inversion of the deleted *AtPROPEP2/1/3* fragment between the gRNA-Pep2 and gRNA-Pep3 induced breakpoints in the *atpropep1-8* $T_2$ line #2-4 (Fig. 1g).

To reconcile above findings with our earlier observation that genomic deletion of *AtPROPEP2/1/3* occurred in the *atpropep1-8* $T_1$ line #2 (Supplementary Fig. 1a), we speculated that the line #2 had been wrongly classified as a homozygous *AtPROPEP2/1/3* deletion allele but instead was a bi-allele containing the deletion of *AtPROPEP2/1/3* on one chromosome but inversion of *AtPROPEP2/1/3* on the other homologous chromosome (Fig. 1e). This mistake was caused by the failure of the tier-2 PCR (using the Fw1/Rev2 primers) to distinguish between TAG inversion and deletion. This type of biallelic mutation was hereafter termed delinver (deletion/inversion bi-allele) for simplicity. To testify our reasoning, we checked the genotype segregation ratio in 96 $T_2$ plants derived from the *atpropep1-8* line #2. We found that 61% (59/96) of its $T_2$ plants carried delinver mutations at the *AtPROPEP2/1/3* locus, while 21% (20/96) harbored homozygous deletion mutations and 18% (17/96) contained homozygous inversion mutations (Supplementary Fig. 3a, b), which roughly fitted the Mendelian segregation ratio. Throughout this study, we applied the modified genotyping PCR scheme containing a tier-3 PCR using a pair of co-aligned primers, namely Rev1 and Rev2 (Fig. 2a), to discriminate homozygous genomic deletion alleles from delinver bi-alleles (Table 1).

The serendipitous observation of delinver mutation at the *AtPROPEP2/1/3* locus in the *atpropep1-8* line #2 prompted us to re-check with closer scrutiny whether delinver mutation had also taken place at the *AtPROPEP8/7/4/5* locus in *atpropep1-8* mutant lines. To this end, we conducted the tier-3 PCR using the Fw1/Fw2 or Rev1/Rev2 primer pairs specific for the *AtPROPEP8/7/4/5* locus (Fig. 2b). Intriguingly, we identified $T_2$ plants from the *atpropep1-8* line #13 (e.g., #13-2), but not from the line #2 or #11, as delinver bi-alleles for

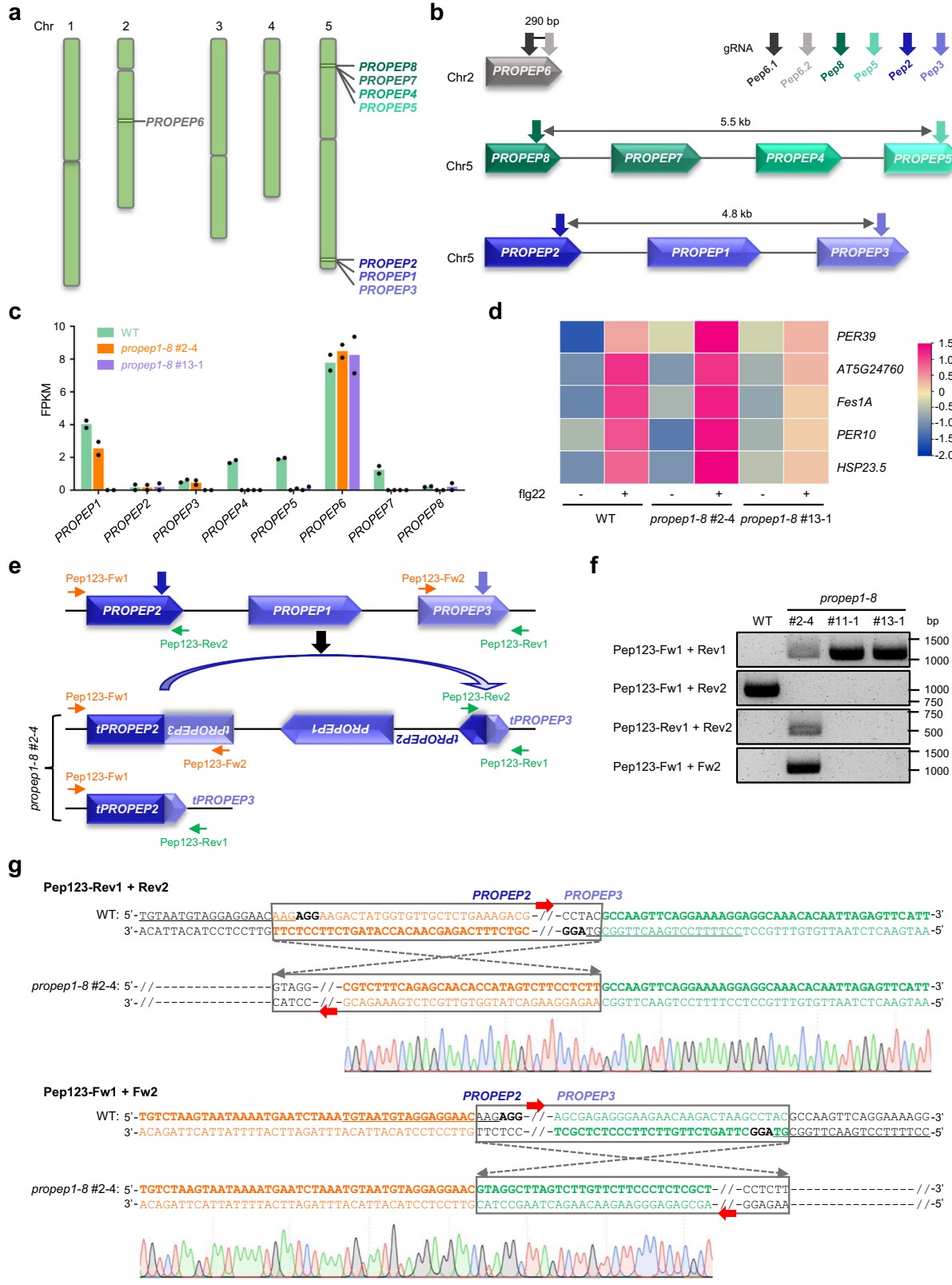

*AtPROPEP8/7/4/5* (Fig. 2b, c). Sanger sequencing of PCR amplicons confirmed that the deleted *AtPROPEP8/7/4/5* fragment was invertedly re-inserted between the gRNA-Pep8 and gRNA-Pep5 induced DSBs in the *atpropep1-8* T$_2$ line #13-2 (Fig. 2d). By inference, the *atpropep1-8* T$_1$ line #13 should have been mistakenly regarded as a homozygous *AtPROPEP8/7/4/5* deletion allele but rather was a delinver bi-allele.

## Delinver mutations are prevalent in targeted TAG deletions in *Arabidopsis*

Since both *AtPROPEP2/1/3* and *AtPROPEP8/7/4/5* are located on chromosome 5, we were wondering whether delinver mutations also take place during multiplexed CRISPR editing at other TAG loci from different chromosomes in *Arabidopsis*. For this purpose, we targeted

**Fig. 1 | Unexpected occurrence of deletion-inversion bi-allele of *AtPROPEP2/1/3* in multiplexed CRISPR editing. a** Chromosomal distribution of *AtPROPEP1-8* in *Arabidopsis*. Chr, chromosome. **b** Paired gRNA strategy for generating *atpropep1-8* octuple mutant plants. **c** RNA-seq analysis revealed that *AtPROPEP1* retains transcription in the presumptive homozygous *atpropep1-8* T₂ line #2-4. Data are shown as mean values of two biological replicates and each dot represents the data of one biological replicate. WT, wild type. **d** AtPep1-responsive genes are normally induced by flg22 in *atpropep1-8* #2-4. Induction fold of each gene under flg22 treatment is indicated by the color scale based on $\log_2$ fold change. Ten-day-old seedlings were treated with or without 100 nM flg22 for 4 h before RNA-seq analysis. **e** Diagram of the deletion-inversion (delinver) bi-allele for *AtPROPEP2/1/3* in *atpropep1-8* #2-4. Primers used for PCR-based genotyping are shown. Fw, forward primer. Rev, reverse primer. t, truncated version. **f** PCR-based genotyping revealed the delinver genotype of *AtPROPEP2/1/3* in *atpropep1-8* #2-4. Experiments were repeated twice with similar results. **g** Sanger sequencing of PCR amplicons using co-aligned primers validated the genomic inversion between gRNA-Pep2 and gRNA-Pep3 induced breakpoints in *atpropep1-8* #2-4. Black bold letters mark PAMs. Target sequences of gRNAs are underlined. Source data are provided as a Source Data file.

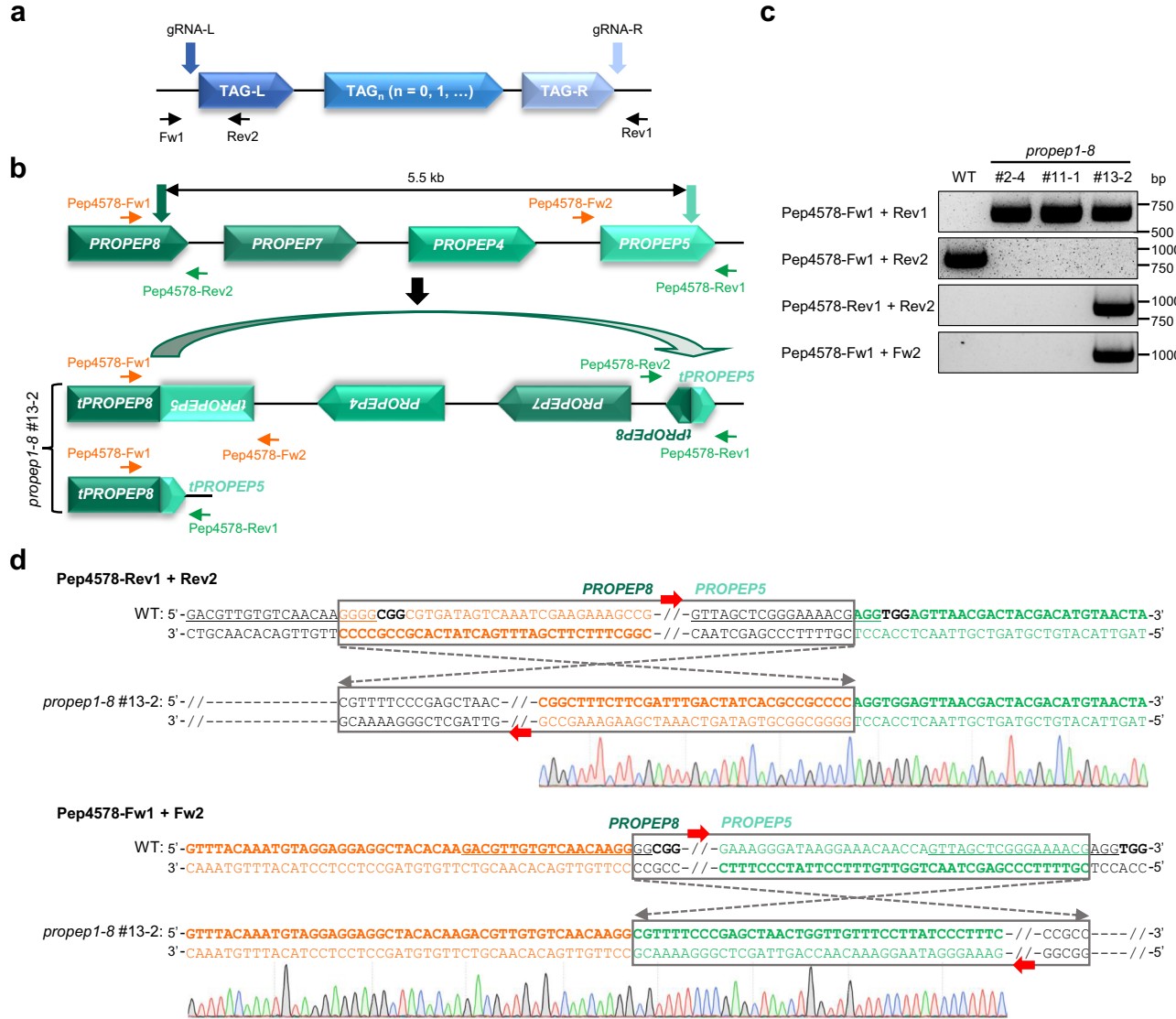

**Fig. 2 | Deletion-inversion bi-allele of *AtPROPEP8/7/4/5* induced by multiplexed CRISPR editing. a** Diagram of a putative TAG locus for CRISPR-mediated deletion with gRNAs and genotyping primers indicated. TAG-L and TAG-R denote the two outermost genes at the TAG locus, respectively. **b** Diagram of the delinver bi-allele for *AtPROPEP8/7/4/5* in the presumptive homozygous *atpropep1-8* T₂ mutant line #13-2. Primers used for PCR-based genotyping are shown. Fw, forward primer. Rev, reverse primer. t, truncated version. **c** PCR-based genotyping revealed the delinver genotype of *AtPROPEP8/7/4/5* in *atpropep1-8* #13-2. Experiments were repeated twice with similar results. **d** Sanger sequencing of PCR amplicons using co-aligned primers validated the genomic inversion between gRNA-Pep5 and gRNA-Pep8 induced breakpoints in *atpropep1-8* #13-2. Black bold letters mark PAMs. Target sequences of gRNAs are underlined. Source data are provided as a Source Data file.

SpCas9 to the TAGs of *AtMC7/6/5/4* encoding four metacaspases on chromosome 1 using the gRNA-MC7 and gRNA-MC4 (Fig. 3a), or to the TAGs of *AtCHI1/2/3/4/5/6* encoding six chitinases on chromosome 2 using the gRNA-CHI1 and gRNA-CHI6 (Fig. 3b). We conducted a primary PCR screen across all transgenic plants to identify TAG deletion-containing alleles using the corresponding Fw1/Rev1 primers and then a secondary PCR screen in the deletion-containing alleles to further identify delinver bi-alleles using the co-aligned primers (Fig. 3a, b). For the TAGs of *AtMC7/6/5/4*, by genotyping 90 transgenic T₁ plants, we identified three plants carrying deletions of the 8.2-kb TAG fragment, in which one plant (33.3%) was a delinver bi-allele (Table 2). For the TAGs of *AtCHI1/2/3/4/5/6*, by genotyping 157

**Table 1 | Results of genotyping PCR and corresponding possible genotypes**

| Summary of genotyping PCR outcomes for different genotypes | | | | | |
|---|---|---|---|---|---|
| PCR | WT | Heterozygous TAG deletion | Homozygous TAG deletion | Delinver bi-allele | |
| Tier 1 (Fw1 + Rev1) | –[a] | +[b] | + | + | |
| Tier 2 (Fw1 + Rev2) | + | + | – | – | Standard genotyping |
| Tier 3 (Rev1 + Rev2) | – | – | – | + | |

[a]The minus symbol indicates negative amplicons.
[b]The plus symbol indicates positive amplicons.

transgenic $T_1$ plants, we detected 30 plants containing deletions of the 18.7-kb TAG fragment, in which 17 plants (56.7%) corresponded to delinver bi-alleles (Table 2). Sanger sequencing of PCR amplicons using the co-aligned primers further verified the inversion of the deleted TAG fragments between the gRNA-MC7 and gRNA-MC4 induced DSBs in the representative *atmc4-7* delinver line #87 (Fig. 3c) and between the gRNA-CHI1 and gRNA-CHI6 induced DSBs in the representative *atchi1-6* delinver line #25 (Fig. 3d). These findings indicate the prevalence of delinver mutations in CRISPR-mediated TAG deletions in *Arabidopsis*.

**Delinver mutations are predominantly associated with efficient chromosomal deletions**

Both the gRNA sequence and chromatin feature of the target site are known as vital determinants for the efficiency of CRISPR-mediated genome editing[43,44]. To obtain clues about the factors that may affect the frequencies of delinver mutations, we extensively evaluated multiplex CRISPR-mediated deletions of the TAGs on chromosome 5 that encode three R proteins, namely AT5G45240, RPS4, and RRS1. By designing three gRNA-Ls (i.e., L1 to L3) to target AT5G45240 and ten gRNA-Rs (i.e., R1 to R10) to target *AtRRS1* (Fig. 4a), we tested 12 gRNA-L/gRNA-R combinations and screened a total of 1,293 transgenic $T_1$ plants for genomic deletions by PCR. The frequencies of TAG deletions were found to range from 0 to 77.8% for different gRNA-L/gRNA-R pairs (Table 2). Among the mutant plants containing 8 ~ 18.4-kb genomic deletions, a secondary PCR screen using two co-aligned primers further revealed that up to 80% (with a median value of 51.3%) of deletion-containing alleles actually corresponded to delinver bi-alleles (Table 2). Sanger sequencing validated the inversions of deleted TAG fragments between the breakpoints induced by corresponding gRNA-L/gRNA-R pairs (Fig. 4b and Supplementary Fig. 4). Overall, the frequencies of delinver mutations appeared to be positively correlated ($R^2 = 0.904$) with those of deletion mutations but unrelated to the deletion sizes (Fig. 4c, d). Moreover, although the gRNAs R3-R6 were purposely designed to target the same chromosomal region with shared chromatin context (Fig. 4a), the gRNA pairs between L1 and R3-R6 gave rise to distinct frequencies of delinver mutations ranging from 0 to 55.6% (Table 2). These results hint that the efficiencies of paired gRNAs, rather than the distance or chromatin features of the target sites, are probably key factors affecting the frequencies of delinver mutations.

**Chromosomal inversion may occur regardless of blunt or staggered ends of DSBs**

SpCas9 creates a blunt-ended DSB at the target site[45], which may facilitate inverted re-insertion of the deleted TAG fragment. By contrast, the LbCpf1 from *Lachnospiraceae bacterium* generates a DSB with a 4 ~ 5 base overhang[46]. This encouraged us to assess whether LbCpf1-induced DSBs can prevent the inversion of TAG deletion fragments. Therefore, we designed a series of CRISPR RNA (crRNA)-Ls (i.e., L1 to L4) and crRNA-Rs (i.e., R1 to R7) for LbCpf1 to target the TAGs of AT5G45240/*AtRPS4*/*AtRRS1* with overlapping or proximal target sites relative to SpCas9 gRNA-Ls and gRNA-Rs (Supplementary Fig. 5a). Unfortunately, in transgenic $T_1$ plants co-expressing *LbCpf1* under an

egg cell-specific promoter[47] with a crRNA-L/crRNA-R pair, we were unable to identify any mutant alleles containing anticipated TAG deletions. This was in agreement with the reported low efficiency of LbCpf1-mediated genome editing in *Arabidopsis*[48,49]. To bypass this issue, we transiently co-expressed crRNA-Ls and crRNA-Rs in a pairwise manner along with *LbCpf1* in *Arabidopsis* protoplasts. TAG inversions could be readily detected by PCR using the co-aligned primers (Supplementary Fig. 5b) and were further validated by amplicon Sanger sequencing (Supplementary Fig. 6). These data suggest that multiplex LbCpf1-induced cohesive ends of DSBs may not be able to block inverted re-insertion of the deleted TAG fragments, though future endeavors in transgenic plants are needed to fully validate this conclusion.

**Delinver mutations occur in rice during multiplexed CRISPR editing**

We next investigated whether multiplexed CRISPR editing could also lead to delinver mutations in rice. For this purpose, we aimed SpCas9 at the TAG locus on chromosome 8 that encodes two Pep phytocytokine receptors plus an unknown protein (LOC_Os08g34630) using four previously reported *OsPEPR*-targeting gRNAs[50] (Fig. 5a). We screened 300 transgenic rice calli by PCR using the primers PEPR1-Fw/PEPR2-Rev (Fig. 5a) and identified 16.3% (49/300) of them carrying >7.2-kb genomic deletions spanning all three genes (Table 3). Among the deletion-containing calli, we conducted PCR with the co-aligned primers PEPR1-Rev/PEPR2-Rev (Fig. 5a) to further pinpoint those harboring chromosomal inversions. As a result, there were 14.3% (7/49) of deletion-containing calli simultaneously carrying large genomic inversions (Table 3). As expected, Sanger sequencing revealed four types of inversions (Supplementary Fig. 7), which were mediated by pairwise combinations of gRNAs targeting *OsPEPR1* or *OsPEPR2*.

Rather than using the paired gRNA strategy to target the two outermost genes at a TAG locus, researchers sometimes utilize multiple gRNAs to target every single gene of TAGs to maximize multiplex editing[51,52]. To examine the occurrence of delinver mutations under such circumstances, we targeted SpCas9 to the TAG locus of *OsPROPEP2/6/5/3/7/4* encoding six rice PROPEP phytocytokines on chromosome 8 by using six gRNAs, with one gRNA for each gene (Fig. 5b). To reduce the complexity in genotyping, we focused on mutant alleles with >14-kb genomic deletions that spanned at least four genes (Fig. 5b), namely between the target sites of gRNA-PEP2/gRNA-PEP3, gRNA-PEP2/gRNA-PEP7, gRNA-PEP2/gRNA-PEP4, gRNA-PEP6/gRNA-PEP4 or gRNA-PEP5/gRNA-PEP4. By PCR-based screening of 303 transgenic rice calli using corresponding Fw/Rev primers flanking possible large deletions, we detected all five types of deletion mutations mediated by the above gRNA pairs, with a frequency of 3.3 ~ 26.7% (Table 3). Among these large deletion-containing calli, we further identified a proportion of delinver calli at 0 ~ 79% by PCR using corresponding co-aligned primers (Table 3). Taken together, these results demonstrate that delinver mutations also occur in rice during multiplexed CRISPR editing.

The gRNA-PEP4, when combined with gRNA-PEP2, gRNA-PEP6 or gRNA-PEP5, gave rise to a delinver/deletion ratio of 12.5%, 0, and 20%, respectively (Table 3). These data suggest that the gRNA-PEP6, but not

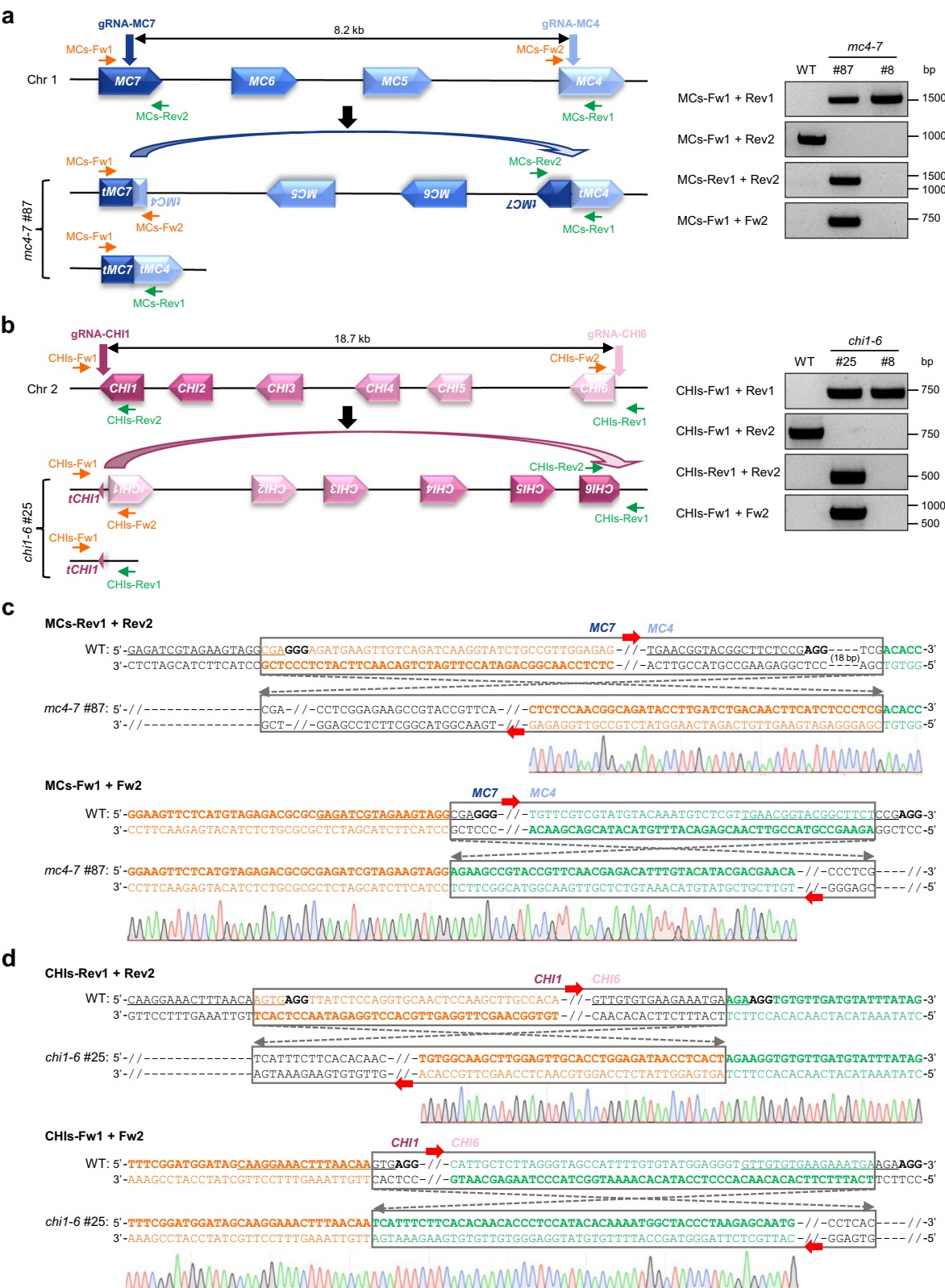

**Fig. 3 | Delinver mutations are prevalent in targeted TAG deletion by multiplexed CRISPR system in *Arabidopsis*. a** PCR-based genotyping revealed the delinver genotype of *AtMC7/6/5/4* in the *atmc4-7* T₁ mutant line #87. Primers used for PCR-based genotyping are shown. Fw, forward primer. Rev, reverse primer. **b** PCR-based genotyping revealed the delinver genotype of *AtCHI1/2/3/4/5/6* in the *atchi1-6* T₁ mutant line #25. In **a** and **b** experiments were repeated twice with similar results. **c, d** Sanger sequencing of PCR amplicons using co-aligned primers validated the inversion of *AtMC7/6/5/4* between gRNA-MC4 and gRNA-MC7 induced breakpoints in *atmc4-7* #87 (**c**) and inversion of *AtCHI1/2/3/4/5/6* between gRNA-CHI1 and gRNA-CHI6 induced breakpoints in *atchi1-6* #25 (**d**). In **c** and **d** black bold letters mark PAMs and target sequences of gRNAs are underlined. Source data are provided as a Source Data file.

**Table 2 | Summary of deletion and delinver mutations mediated by different gRNA pairs at different TAG loci in *Arabidopsis***

| TAGs | gRNAs | Deletion size (kb) | Transgenic plants | Plants with deletions | Deletion frequency (%) | Plants with delinvers | Delinver frequency (%) | Delinver/deletion ratio (%) |
|---|---|---|---|---|---|---|---|---|
| *AtMC7/6/5/4* | gRNA-MC7 gRNA-MC4 | 8.2 | 90 | 3 | 3.3 | 1 | 1.1 | 33.3 |
| *AtCHI1/2/3/4/5/6* | gRNA-CHI1 gRNA-CHI6 | 18.7 | 157 | 30 | 19.1 | 17 | 10.8 | 56.7 |
| AT5G45240/ *AtRPS4*/ *AtRRS1* | L1 + R1 | 18.4 | 147 | 113 | 76.9 | 83 | 56.5 | 73.5 |
| | L1 + R2 | 18.3 | 8 | 5 | 62.5 | 2 | 25.0 | 40.0 |
| | L1 + R3 | 18.2 | 32 | 0 | 0 | 0 | 0 | 0 |
| | L1 + R4 | 18.2 | 383 | 270 | 70.5 | 203 | 53.0 | 75.2 |
| | L1 + R5 | 18.2 | 10 | 2 | 20.0 | 1 | 10.0 | 50.0 |
| | L1 + R6 | 18.2 | 18 | 14 | 77.8 | 10 | 55.6 | 71.4 |
| | L1 + R7 | 18.1 | 218 | 132 | 60.6 | 90 | 41.3 | 68.2 |
| | L1 + R8 | 17.9 | 101 | 6 | 5.9 | 0 | 0 | 0 |
| | L2 + R9 | 14.3 | 24 | 15 | 62.5 | 12 | 50.0 | 80.0 |
| | L2 + R10 | 11.2 | 98 | 39 | 39.8 | 20 | 20.4 | 51.3 |
| | L3 + R9 | 11 | 46 | 13 | 28.3 | 2 | 4.3 | 15.4 |
| | L3 + R10 | 8 | 208 | 24 | 11.5 | 6 | 2.9 | 25.0 |

gRNA-PEP4, is responsible for the absence of delinver mutations in the case of gRNA-PEP4/gRNA-PEP6 pair. Meanwhile, the gRNA-PEP2/gRNA-PEP3 pair led to substantial deletion and delinver mutations (Table 3), implying that both gRNAs are highly active. Therefore, we wondered how the gRNA-PEP2/gRNA-PEP6 pair would perform. Notably, this gRNA pair generated 20 transgenic rice calli containing ~15-kb genomic deletions, but none of them harbored delinver mutations (Table 3). This result again indicates that the gRNA-PEP6 somehow hinders the formation of delinver mutations.

**Delinver mutations are not specific to a TAG locus**
Finally, we asked whether high levels of delinver mutations could occur at a non-TAG locus. To this end, we designed two gRNAs (i.e., L4 and L5) targeting AT5G45290 in *Arabidopsis* (Fig. 6a). By pairwise combining gRNA-L4 or gRNA-L5 with gRNA-R3 or gRNA-R4 targeting *AtRRS1* for CRISPR-mediated genomic deletion, we attempted to evaluate whether deleting the genomic region adjacent to the TAGs of AT5G45240/*AtRPS4*/*AtRRS1* with a similar size (i.e., ~18.2 kb) could result in delinver mutations or not. The gRNA-R3 and gRNA-R4 were selected because these two overlapping gRNAs exhibited contrasting efficiencies (that is, 0 or 53%) in inducing delinver mutations when combined with the gRNA-L1 targeting AT5G45240 (Table 2). By genotyping a total of 320 transgenic plants, we observed genomic deletions mediated by all four gRNA pairs, with a frequency of 5.3 - 37.5% (Table 4). Among the deletion-containing plants, we further identified delinver bi-alleles with a proportion of 0 - 70.4% (Table 4, Fig. 6b and Supplementary Fig. 8). These results suggest that delinver mutations also take place at high frequencies at genomic non-TAG loci during multiplexed CRISPR editing.

Similar to TAGs, many genes regulating the biosynthesis of secondary metabolites are organized as operon-like gene clusters in plant genomes[53]. We selected a ~32-kb metabolic gene cluster (i.e., AT5G47980, *AtTHAD*, *AtTHAH*, and *AtTHAS*) required for triterpene biosynthesis in *Arabidopsis*[54] for CRISPR-mediated genomic deletion using two different gRNA pairs (Fig. 6c). As a result, 56.3% (45/80) of transgenic plants expressing the gRNA-L1/gRNA-R1 pair were characterized to contain genomic deletions, in which 66.7% (30/45) corresponded to delinver bi-alleles (Table 4 and Supplementary Fig. 9). Meanwhile, 7.4% (7/95) of transgenic plants expressing the gRNA-L2/gRNA-R2 pair were deletion-containing mutants, in which 42.9% (3/7) carried delinver mutations (Table 4 and Fig. 6d). These findings further substantiate the notion that delinver mutations are not specific to a

TAG locus. In addition, the data obtained from the two non-TAG loci again reflect an overall trend that the frequencies of delinver mutations are positively correlated ($R^2 = 0.977$) with those of deletion mutations but not with the deletion sizes (Fig. 6e, f).

## Discussion
In plants, in-depth functional annotation of TAGs has been long impeded by their genetic redundancy and chromosomal linkage[1]. The multiplex CRISPR system has nowadays provided a powerful tool for efficiently generating TAG knockout alleles. Using the paired gRNA strategy, where two gRNAs are individually targeted to the outermost genes at a TAG locus, TAG deletion between the CRISPR-induced DSBs can be promoted[17–26]. Subsequently, a standard two-tier PCR screen allows easy identification of TAG knockout alleles. However, the limit of PCR-based genotyping is that it critically depends on foreseeable editing outcomes. Therefore, unintended genomic alterations in edited plants are prone to evading common PCR-based genotyping.

In this study, we stumbled across such unexpected genomic changes when employing the paired gRNA strategy for targeted deletion of two TAG loci encoding *Arabidopsis* PROPEPs. Three presumptive *atpropep1-8* octuple mutant lines were initially identified in T₁ generation by the two-tier PCR-based genotyping, whereas two of them happened to carry intended TAG deletions on one chromosome but unintended inversion of the deleted TAG fragment of either *AtPROPEP2/1/3* (line #2, Fig. 1) or *AtPROPEP8/7/4/5* (line #13, Fig. 2) on the other homologous chromosome. Although several studies have reported that CRISPR-mediated DSBs can stimulate various types of unexpected genomic rearrangements in mammalian and plant cells[35–40], such deletion-inversion (delinver) bi-alleles have not been documented. Importantly, we observed delinver mutations at multiple TAG loci on different chromosomes in *Arabidopsis* (Table 2) and rice (Table 3), where up to ~80% of TAG deletion alleles corresponded to delinver bi-alleles. Moreover, comparable levels of delinver mutations were also detected at genomic non-TAG loci (Table 4). In total, out of 31 pairs of gRNAs that induced large genomic deletions in this study, 27 gave rise to delinver mutations at varying frequencies (Figs. 1e and 2b and Tables 2–4). Collectively, our findings indicate the prevalence of delinver mutations induced by multiplexed CRISPR editing in plants.

We noted that high frequencies of delinver mutations were often associated with efficient genomic deletions (Figs. 4c and 6e), whereas earlier studies attempting to generate targeted chromosomal inversions only observed a rather low frequency of inversion events

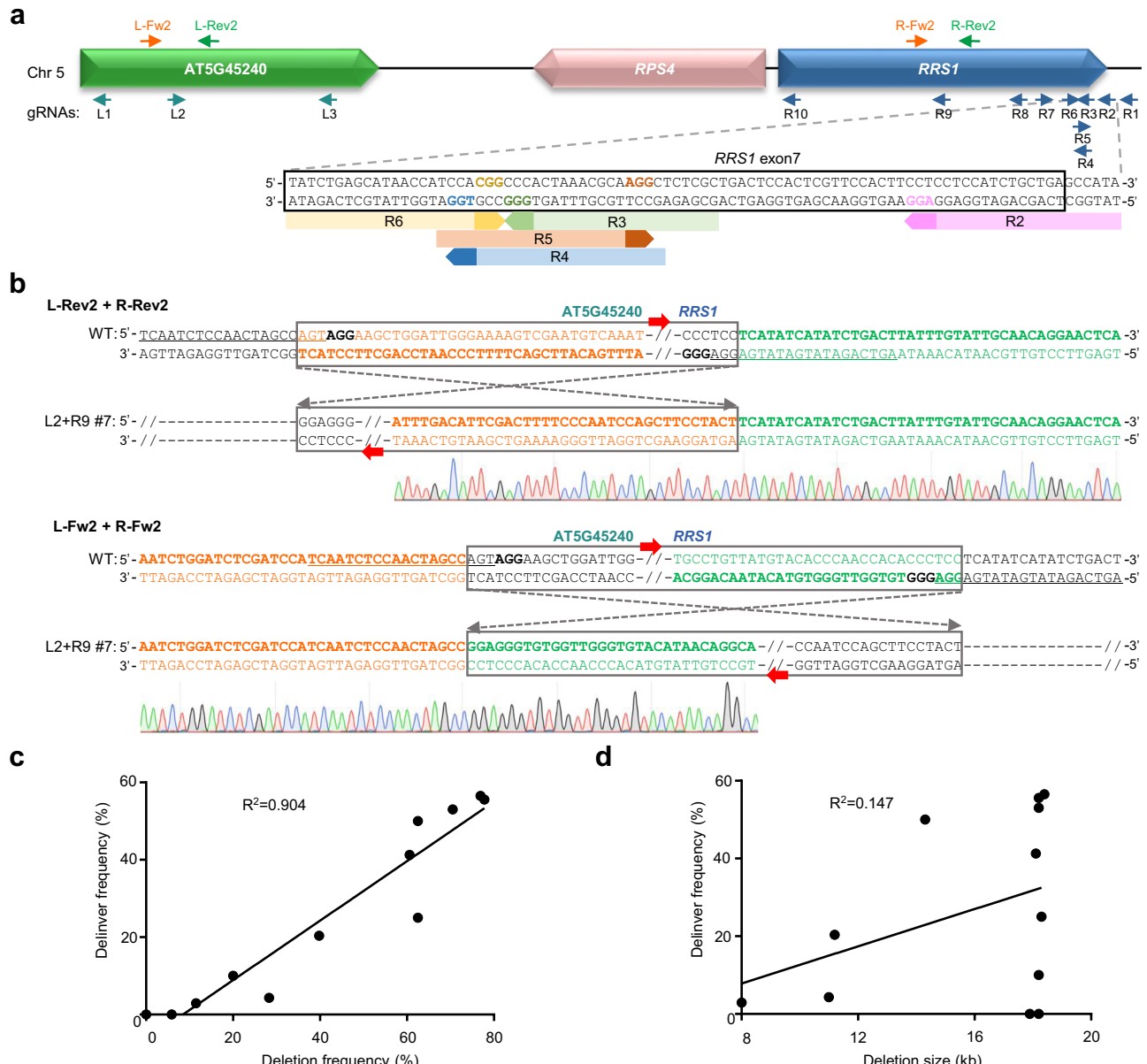

**Fig. 4 | Frequencies of delinver mutations are positively correlated with those of deletion mutations. a** Diagram of the AT5G45240/*AtRPS4*/*AtRRS1* locus. Three gRNA-Ls (L1 to L3) targeting AT5G45240 and ten gRNA-Rs (R1 to R10) targeting *AtRRS1* were designed for testing the frequencies of delinver mutations mediated by different gRNA-L/gRNA-R pairs. gRNAs targeting to antisense or sense strands are indicated by leftwards and rightwards arrows, respectively. Note that the gRNAs R2-R6 were targeted to the same region of *AtRRS1*. Colored letters mark PAMs. L-Fw2/R-Fw2 and L-Rev2/R-Rev2 are examples of co-aligned primer pairs, which were used for detecting genomic inversions between gRNA-L2 and gRNA-R9 induced cut sites. **b** Representative Sanger sequencing results of PCR amplicons using co-aligned primers validated the inversion between gRNA-L2 and gRNA-R9 induced breakpoints in the L2 + R9 mutant line #7. Black bold letters mark PAMs and target sequences of gRNAs are underlined. **c** Frequencies of delinver mutations are positively correlated with those of deletion mutations. **d** Frequencies of delinver mutations are unrelated to the deletion sizes. In **c** and **d** each dot represents the data of one gRNA pair and the Pearson correlation was calculated using the GraphPad Prism algorithm.

alone[27,28,55]. These findings together provide a clue that the genomic deletion on one chromosome may promote the inversion of the deleted segment on the other homologous chromosome. It is possible that the two homologous chromosomes align with each other in close proximity, so simultaneous deletion of two homologous segments gives each segment five opportunities, to be re-ligated in WT or reverse orientation back to its own chromosome or to the other homologous chromosome or not to be re-ligated. As long as the deleted segment is lost from one chromosome without re-ligation, the inversion on the other homologous chromosome might be stimulated. From an evolutionary viewpoint, such a mechanism can minimize the detrimental impact of genomic deletion on organismal fitness and meanwhile

increase genetic variations in progeny to facilitate natural selection-based adaptation[56,57].

In addition to the genomic deletion efficiency, there are other parameters affecting the occurrence of delinver mutations. For instance, at the rice TAG locus encoding six OsPROPEPs, the gRNA-PEP6 in combination with either the gRNA-PEP2 or gRNA-PEP4 resulted in a total of 32 transgenic calli carrying genomic deletions. However, none of them appeared to be delinver bi-alleles (Table 3). At the *Arabidopsis* TAG locus of AT5G45240/*AtRPS4*/*AtRRS1*, the gRNA-L1/gRNA-R3 pair failed to produce any genomic deletions, whereas the gRNA-L1/gRNA-R4 pair worked efficiently (Table 2), which indicated the ineffectiveness of gRNA-R3. In line with this

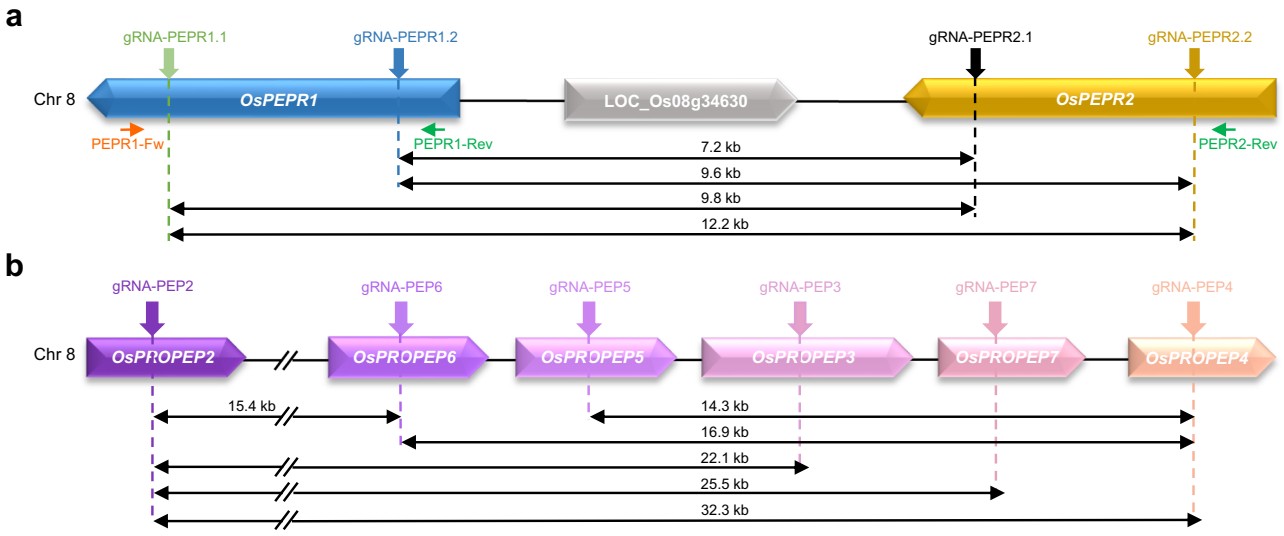

**Fig. 5 | Delinver mutations occur in targeted TAG deletion by multiplex CRISPR system in rice. a** Diagram of the *OsPEPR1/OsPEPR2* locus with gRNAs and genotyping primers indicated. Fw, forward primer. Rev, reverse primer. **b** Diagram of the *OsPROPEP2/6/5/3/7/4* locus with gRNAs indicated.

**Table 3 | Summary of deletion and delinver mutations mediated by different gRNA pairs at different TAG loci in rice**

| TAGs | gRNAs | Deletion size (kb) | Transgenic calli | Calli with deletions | Deletion frequency (%) | Calli with delinvers | Delinver frequency (%) | Delinver/deletion ratio (%) |
|---|---|---|---|---|---|---|---|---|
| *OsPEPR1/ LOC_Os08g34630/ OsPEPR2* | gRNA-PEPR1.1 gRNA-PEPR1.2 gRNA-PEPR2.1 gRNA-PEPR2.2 | 7.2 ~ 12.2 | 300 | 49 | 16.3 | 7 | 2.3 | 14.3 |
| *OsPROPEP2/6/5/ 3/7/4* | gRNA-PEP2 gRNA-PEP3 | 22.1 | 303 | 81 | 26.7 | 64 | 21.1 | 79.0 |
| | gRNA-PEP2 gRNA-PEP7 | 25.5 | | 13 | 4.3 | 1 | 0.3 | 7.7 |
| | gRNA-PEP2 gRNA-PEP4 | 32.3 | | 32 | 10.6 | 4 | 1.3 | 12.5 |
| | gRNA-PEP6 gRNA-PEP4 | 16.9 | | 12 | 3.9 | 0 | 0 | 0 |
| | gRNA-PEP5 gRNA-PEP4 | 14.3 | | 10 | 3.3 | 2 | 0.7 | 20.0 |
| | gRNA-PEP2 gRNA-PEP6 | 15.4 | | 20 | 6.6 | 0 | 0 | 0 |

speculation, the gRNA-R3 also cooperated poorly with the gRNA-L4 to mediate genomic deletion of the adjacent non-TAG region (Table 4). However, when the gRNA-R3 was paired with the gRNA-L5, 37.5% (39/104) of transgenic plants were found to carry genomic deletions, in which 21 plants (53.8%) represented delinver bi-alleles (Table 4). These findings underscore the unpredictable complexity that successful genomic deletions do not always lead to delinver mutations and one gRNA can mediate delinver mutations at distinct frequencies when paired with different gRNAs.

A lesson with important practical relevance in this study is that inverted re-insertions must be cautiously considered in targeted TAG deletion by multiplexed CRISPR editing in plants. In routine two-tier PCR-based genotyping, delinver bi-alleles can be wrongly identified as homozygous TAG knockout alleles (Table 1), resulting in mis-interpretation of experimental data obtained (Fig. 1c, d). Moreover, such oversight can lead to the failure of recognizing the genetic het-erogeneity of progenies produced from delinver bi-alleles (Supple-mentary Fig. 3b), which further confounds follow-up research based on those genetic materials. To avoid this problem, we suggest conducting a three-tier PCR-based genotyping (Table 1) that enables the dis-crimination between homozygous TAG deletion alleles and delinver bi-alleles, especially when a high frequency of TAG deletions have been

detected. Considering that we still have much to learn about the unexpected chromosomal rearrangements associated with multi-plexed CRISPR editing, whole genome sequencing will provide an unbiased picture of what really has happened in a genome-edited plant[40]. Finally, given the prevalence of TAGs in the human genome[58], it is worthwhile to investigate whether multiplexed CRISPR editing of human TAGs in functional studies will also encounter this hidden genotype issue at high frequencies.

## Methods

### Plant materials and growth conditions

The *Arabidopsis thaliana* ecotype Col-0 and rice (*Oryza sativa*) cultivar Zhonghua 11 (ZH11) plants were used as wild-type plants in this study. Transgenic *Arabidopsis* seeds were screened on 1/2 × Murashige and Skoog (MS) solid medium containing 0.05% MES, 0.5% sucrose, 0.8% agar, and 25 mg/L hygromycin. After stratification at 4 °C for 2 days, seeds were geminated in a plant growth chamber (Ningbo Saifu, China) under photoperiods of 16 h light (75 μmol m⁻² s⁻¹) at 23 °C and 8 h dark at 20 °C. The resistant plants were transferred to Jiffy soil (Jiffy Group, Netherlands) and grown in a plant growth room under photoperiods of 12 h light at 23 °C and 12 h dark at 21 °C with humidity main-tained at 65%.

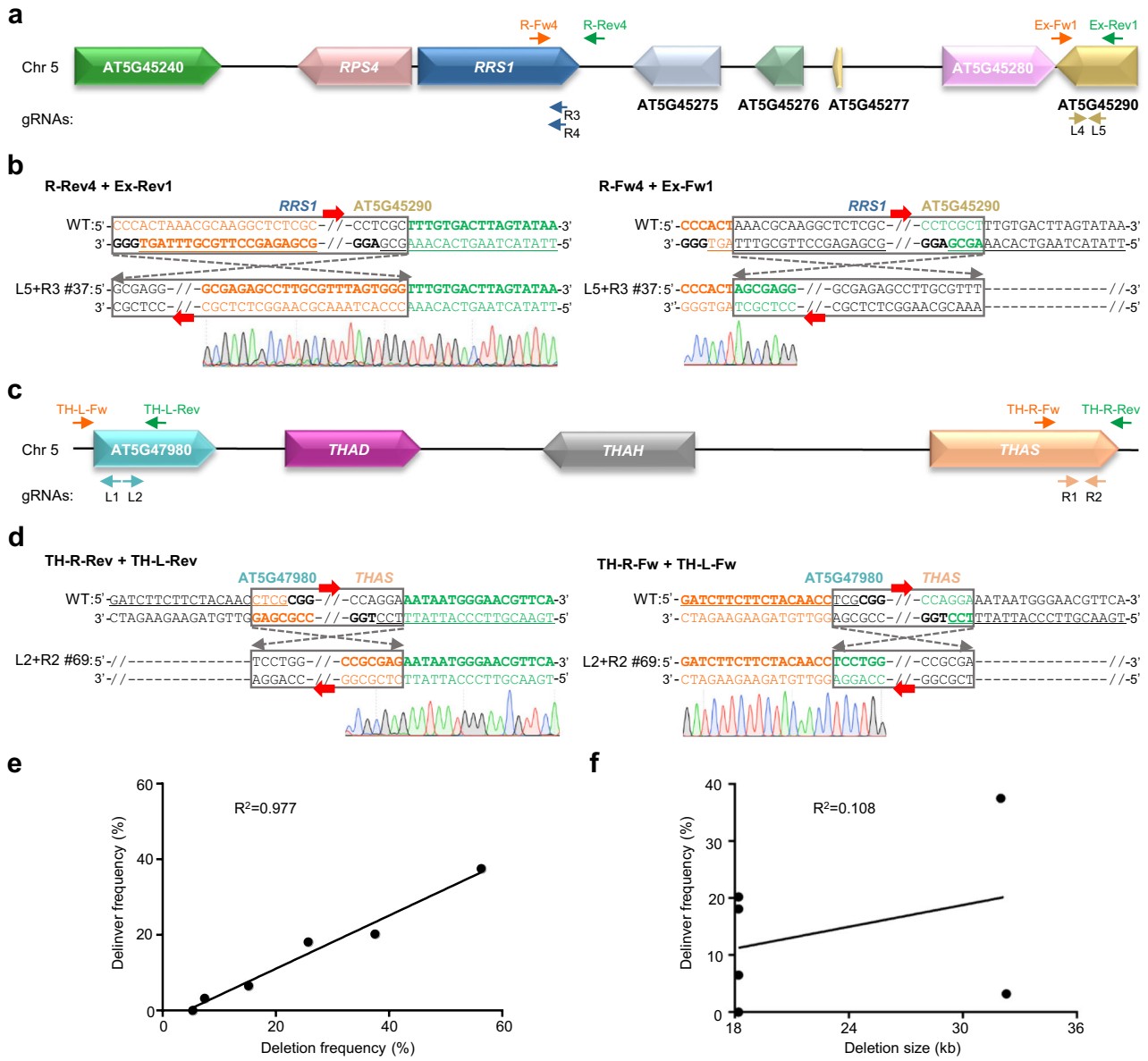

**Fig. 6 | Delinver mutations also occur at genomic non-TAG loci. a** Diagram of the targeted genomic non-TAG locus adjacent to the TAG locus of AT5G45240/ *AtRPS4/AtRRS1* with gRNAs and genotyping primers indicated. Fw, forward primer. Rev, reverse primer. **b** Representative Sanger sequencing results of PCR amplicons using co-aligned primers validated the inversion between gRNA-L5 and gRNA-R3 induced breakpoints in the L5 + R3 mutant line #37. Black bold letters mark PAMs and target sequences of gRNAs are underlined. **c** Diagram of the non-TAG locus of AT5G47980/*AtTHAD/AtTHAH/AtTHAS* with gRNAs and genotyping primers indicated. **d** Representative Sanger sequencing results of PCR amplicons using co-aligned primers validated the inversion between gRNA-L2 and gRNA-R2 induced breakpoints in the L2 + R2 mutant line #69. Black bold letters mark PAMs and target sequences of gRNAs are underlined. **e** Frequencies of delinver mutations are positively correlated with those of deletion mutations. **f** Frequencies of delinver mutations are unrelated to the deletion sizes. In **e** and **f** each dot represents the data of one gRNA pair and the Pearson correlation was calculated using the GraphPad Prism algorithm.

**Table 4 | Summary of deletion and delinver mutations mediated by different gRNA pairs at different non-TAG loci in *Arabidopsis***

| Non-TAGs | gRNAs | Deletion size (kb) | Transgenic plants | Plants with deletion | Deletion frequency (%) | Plants with delinver | Delinver frequency (%) | Delinver/deletion ratio (%) |
|---|---|---|---|---|---|---|---|---|
| *AtRRS1*/ AT5G45275/ AT5G45276/ AT5G45277/ AT5G45280/ AT5G45290 | L4 + R3 | 18.2 | 19 | 1 | 5.3 | 0 | 0 | 0 |
| | L5 + R3 | 18.2 | 104 | 39 | 37.5 | 21 | 20.2 | 53.8 |
| | L4 + R4 | 18.2 | 92 | 14 | 15.2 | 6 | 6.5 | 42.9 |
| | L5 + R4 | 18.2 | 105 | 27 | 25.7 | 19 | 18.1 | 70.4 |
| AT5G47980/ *AtTHAD/ AtTHAH/ AtTHAS* | L1 + R1 | 32 | 80 | 45 | 56.3 | 30 | 37.5 | 66.7 |
| | L2 + R2 | 32.3 | 95 | 7 | 7.4 | 3 | 3.2 | 42.9 |

## Plasmid construction

All gRNAs used in this work were designed by the CRISPR-GE webserver[59] (skl.scau.edu.cn) unless otherwise specified. The binary plasmids expressing gRNAs for knocking out *AtMC7/6/5/4* or *OsPEPR1/OsPEPR2* have been reported elsewhere[25,50]. Other gRNAs used in *Arabidopsis* were assembled as multiple *AtU6-26:gRNA:TTTTTT* expression cassettes by overlapping PCR, which were inserted into the *Hind*III and *Nco*I sites of the pHEE401E plasmid expressing SpCas9 under the egg cell-specific EC1.2 promotor[47]. The gRNAs used for knocking out *OsPROPEP1-7* were ligated into the *Bsa*I-digested gRNA intermediate plasmids, in which the promotors *OsU6a/OsU6a/OsU6b/OsU6b/OsU6c/OsU6c/OsU3m* were used respectively to control gRNA expression. The gRNA expression cassettes were then amplified by PCR and inserted into the pYLCRISPR/Cas9 binary plasmid through Golden Gate ligation according to a detailed protocol[60]. The *UBQ10:crRNA-HSP* expression cassettes were assembled using the oligo phosphorylation and annealing method and then inserted into the *Bsa*I site of the pUC119 plasmid for protoplast expression. The *LbCpf1* coding sequence was cloned into the HBT vector containing the constitutive *35SPPDK* promoter using the ClonExpress II One Step Cloning Kit (Vazyme, China) for protoplast expression. The target sequences of gRNAs and crRNAs are listed in Supplementary Data 1.

## Generation of transgenic plants or calli

The binary plasmids co-expressing SpCas9 and gRNAs were introduced by electroporation into *Agrobacterium tumefaciens* strain GV3101 for *Arabidopsis* transformation or strain EHA105 for rice transformation. The *Agrobacterium* cells carrying appropriate binary plasmids were used to transform *Arabidopsis* through the floral dip method[61] or rice embryogenic calli by following a detailed protocol[62]. Briefly, the developing floral tissues of *Arabidopsis* wild-type plants were submerged in the *Agrobacterium* suspension solution containing 5% sucrose and 0.05% Silwet L-77 for 10 seconds and then covered for one day to maintain high humidity before being cultivated under normal conditions. Generation of transgenic rice calli involved the processes of induction of rice embryogenic calli from the scutella of mature seeds, infection of calli with *Agrobacterium*, co-cultivation of *Agrobacterium* and calli, followed by two rounds of selection for hygromycin-resistant calli.

## Genomic DNA extraction

For *Arabidopsis* plants, the crude genomic DNA (gDNA) extracts were obtained by homogenizing 3 leaves from a single plant in TKE buffer (100 mM Tris-HCl, pH 9.5, 1 M KCl, 10 mM EDTA) and then incubated at 70 °C for 30 min. The 10-fold diluted gDNA extract was used as PCR template for genotyping. For *Arabidopsis* protoplasts and rice calli, the gDNA was extracted by the standard CTAB method. Briefly, the protoplasts and calli were frozen and ground into homogenates, which were mixed up with CTAB extraction solution (100 mM Tris-HCl, pH 8.0, 20 mM EDTA, pH 8.0, 1.4 M NaCl, 2% CTAB, 1% PVP40000) and then incubated at 65 °C for 1 h. After the cell lysates being cooled down to room temperature, the isovolumetric mixture of phenol: chloroform: isoamyl alcohol (25:24:1, v/v) was added. The suspension was incubated at room temperature for 5 min and centrifuged at $16,300 \times g$ for 5 min. The supernatant was gently mixed with isovolumetric-cooled isopropanol and incubated at −20 °C overnight. After centrifugation at $16,300 \times g$ for 10 min, the DNA sediment was washed twice using 75% ethanol. Sterile water was added to dissolve the dried DNA sediment.

## Genotyping of transgenic plants and calli

PCR-based genotyping was conducted using 2 × Rapid Taq Master Mix (Vazyme, China). The extension efficiency of this Taq DNA polymerase is 15 sec/kb. To screen for TAG deletions or inversions, the extension time was set according to the expected sizes of PCR amplicons. PCR amplicons were solved by agarose gel electrophoresis and were visualized using the G:Box F3 gel imaging system (Syngene) controlled by the GeneSys image capture software (version 1.3.1). The amplicon-containing gel strips were excised and subjected to Sanger sequencing on an Applied BiosystemsTM 3730XL platform by the Sangon Biotech company (Shanghai, China). Sanger sequencing data were analyzed using the Snapgene software (version 7.0). The primers used for genotyping PCR in this work are listed in Supplementary Data 2.

## MAMP treatment

The *atpropep1-8* lines #2-4 and #13-1 and wild-type plants were grown vertically on the 1/2 × MS solid medium for 10 days. Seedlings were transferred to a 6-well culture plate containing 1 mL 1/2 × MS liquid medium and incubated in a plant growth chamber overnight. The medium was then removed and the seedlings were treated with 100 nM flg22 or sterile water as mock. After 4 h treatment, the seedlings were frozen with liquid nitrogen and stored at −80 °C for RNA isolation. Each sample contained 9 seedlings with two biological replicates.

## RNA isolation and RNA-seq

Total RNA was extracted using the RNAiso Plus reagent (Takara Bio, Japan) according to the manufacturer's instructions. For RNA-seq analysis, a total amount of 1 μg RNA per sample was used for library construction. RNA-seq was performed on an Illumina NovaSeq 6000 platform by the Biomarker Technologies company (Beijing, China). The raw reads of RNA-seq were processed with a bioinformatic pipeline tool, the BMKCloud online platform (www.biocloud.net). Clean reads were mapped to the Arabidopsis TAIR10 genome assembly using the HISTAT2 tool (version 2.2.0)[63]. Differential expression analysis was performed using the edgeR tool (version 3.6.2)[64].

## TAIL-PCR

TAIL-PCR was performed by following a detailed protocol[65]. Briefly, pre-amplification was performed using gDNA from *atpropep1-8* #2-4 plants as templates and the primers AC0, RB-0a (or LB-0a), and mLADs (mLAD1 to mLAD4). The 50-fold diluted products of the pre-amplification were used as templates for primary TAIL-PCR, which was performed using nested specific primers RB-1a (or LB-1a) and AC1. The 20-fold diluted products of primary TAIL-PCR were used as a template for secondary TAIL-PCR using nested primers RB-2a (or LB-2a) and AC1. The primers LB-0a, LB-1a and LB-2a specifically matched the antisense strand of *AtPROPEP1* to uncover the upstream sequence of *AtPROPEP1*, while the primers RB-0a, RB-1a and RB-2a specifically matched the sense strand of *AtPROPEP1* to uncover the downstream sequence of *AtPROPEP1*. PCR amplicons were separated by agarose gel electrophoresis and the corresponding gel strips were excised and sent for Sanger sequencing. The primers used for TAIL-PCR are listed in Supplementary Data 2.

## Protoplast isolation and transfection

Leaves of four-weak-old *Arabidopsis* plants were used for protoplast isolation and transfection as previously described[66]. Briefly, detached leaves were cut into 0.5 mm stripes and were digested in enzyme solution (1.5% Cellulase R10, 0.4% macerozyme R10, 0.4 M mannitol, 20 mM MES, pH5.7, 20 mM KCl, 10 mM CaCl$_2$, 0.1% BSA) at room temperature for 3 h. Equal volume of W5 solution (154 mM NaCl, 125 mM CaCl$_2$, 5 mM KCl, 2 mM MES, pH 5.7) was added to the digestion mixture. The cells were filtered with FALCON cell strainer and collected by centrifugation at $100 \times g$ for 2 min. Protoplasts were resuspended with W5 solution and rested on ice for 30 min. After centrifugation, the protoplasts were resuspended in MMG solution (0.4 M mannitol, 15 mM MgCl$_2$, 4 mM MES, pH 5.7) to a concentration of $2 \times 10^5$ cells per mL. Transfection was carried out by mixing 400 μL protoplasts with 40 μL plasmid DNA (2 μg/μL) expressing *LbCpf1* and paired crRNAs and 440 μL PEG solution (40% PEG4000, 0.2 M

mannitol, 0.1 M CaCl₂). The transfection mixture was incubated at room temperature for 5 min and then the transfection was stopped by adding 1.6 mL W5 solution. Transfected cells were pelleted by centrifugation at $100 \times g$ for 2 min and resuspended in 200 μL W5 solution. Cell suspension was transferred to 2 mL WI solution (0.5 M mannitol, 20 mM KCl, 4 mM MES pH 5.7) in a 6-well culture plate. Transfected protoplasts were incubated at room temperature for 18 h before gDNA was extracted for genome editing analysis.

## Reporting summary
Further information on research design is available in the Nature Portfolio Reporting Summary linked to this article.

## Data availability
The RNA-seq data generated in this study have been deposited in the Sequence Read Archive (SRA) database under accession PRJNA923581. The CRISPR target sequences and PCR primer sequences are included in the Supplementary Data files. Source data are provided with this paper.

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

## Acknowledgements

This work was supported by the National Key Research and Development Program of China grant 2019YFA0906202 to J.-F.L. and the National Natural Science Foundation of China grant 31900305 and the Natural Science Foundation of Guangdong Province grant 2020A1515010465 to F.-Z.W. The authors thank Qi-Jun Chen for providing the pHEE401E construct and Ke-Jian Wang for the LbCas12a construct.

## Author contributions

J.L., C.L., and Y.L. conducted the experiments. J.L. and F.-Z.W. analyzed the data. F.-Z.W. and J.-F.L. wrote the manuscript with input from all authors. J.-F.L. conceived and supervised the study.

## Competing interests

The authors declare no competing interests.
