## [Peer Review File · Nature Communications]

Hidden prevalence of deletion-inversion bi-alleles in CRISPR-mediated deletions of tandemly arrayed genes in plantsReviewers' Comments:

Reviewer #1:

Remarks to the Author:

Reading the manuscript of Liu et al I became more and more surprised by the assessments of the authors. The reason for this is that I find the results by far less surprising as the authors and I also do not agree with a row of their conclusions. I do not agree at all with the conclusion "this is the first report about unintended chromosomal rearrangements induced by CRISPR in plants" (line 317) See e.g. as one case of many the retraction published in Nature Communications <https://doi.org/10.1038/s41467-021-24195-5>, where a large unwanted deletion was found and not a conversion as published before.

I also think that the more anecdotal findings of the authors that in tandemly arranged genes (TAG) inversions combined with deletions can occur at high frequencies is much less surprising than the author want to make us believe. I guess they were so surprised as they did not take this option into account at all when they first analyzed mutation induction in multiple genes of a tandem array. Obviously, in TAGs such events are more difficult to detect and the paper would well deserve as a note of caution publication in more specialized journal. However, I do see no applicable technical advance nor major new insights in the nature of DSB repair.

Indeed, several published reports showed already that not only deletions but also inversions can occur with high frequencies in plants (see e.g. the Schmidt et al., 2019 citation in the manuscript). The authors speculate that inversion efficiency is especially high within TAGs (line 342) but neither are they able to sustain this hypothesis by comparative experiments nor do they suggest a molecular model to explain this feature

The authors also state that they found no correlation between inversion sizes and efficiencies (line 222 following). I am not surprised as the authors have to take the cutting efficiencies of the individual sgRNAs into account, which they don't mention. Indeed, they report on a correlation with simple deletion efficiencies, which is absolutely in line with cutting efficiencies as decisive factor. Obviously, the more efficient you cut, the higher the chances are to cut not only both sites within a chromosome but also both homologues and obtain besides simple deletions and inversion, deletions in both alleles as well as deletions combined with inversions. Looking at their rice experiments it becomes clear that there is nothing one has to be surprised They find deletions in 49 of 300 cases, which is high but not exceptional: From these 49, 7 contain besides the deletion an inversion. But how many of the pure deletions have both alleles deleted? Most probably much more than inversions. This all absolutely within the ranges reported before for these rearrangements by others and by no means exceptional.

Reviewer #2:

Remarks to the Author:

Due to the easiness of assembly and expression of multiple guide RNAs, multiplexed genome editing by CRISPR systems has been widely used in plant research and applications. However, the potential unexpected consequences of multiplexed editing by CRISPR have not been fully addressed in plants. In this study, Liu et. al made stunning discovery the high frequency deletion-inversion bi-alleles when targeting tandemly arrayed genes (TAGs) for deletion by CRISPR-Cas9 in both Arabidopsis and rice plants. The authors demonstrated their discovery by targeting different TAGs on different chromosomes, and the data is quite convincing. Overall, I found this manuscript very interesting and well written. All the experiments were carefully designed with rigorous analysis. The conclusions are generally sound. I don't have any major criticisms. Rather, I feel the discovery and novelty are quite significant to the field genome editing beyond plants. It is appropriate to consider its publishing in Nature Communications.

Minor points:

1. The authors mentioned in multiple occasions (e.g., lines 84-85, 307-308, and 316-317) that no studies have investigated the unintended chromosomal rearrangements induced by CRISPR in plants.

While I echo the general point by the authors on this surprisingly understudied topic (given the importance of the issue), this claim however is not true anymore. Recently, Zhang et. al reported a genome-wide investigation on off-target effects by multiplexed CRISPR-Cas12a editing in rice (<https://doi.org/10.1002/tpg2.20266>). They discovered unexpected chromosomal rearrangements such as large deletion, duplication, and translocation.

2. For the Cas12a (Cpf1) editing experiment, since it is done in bulked cells as Arabidopsis protoplasts, it is not possible to tell whether there are, or which are delinver events. The authors demonstrated inversion can occur with staggered end cutting, which was also demonstrated with ZFNs in Arabidopsis (Qi, et. al G3, 2013). However, their conclusion as stated in lines 332-333 is not accurate because the statement implies that the authors proved delinver events with Cpf1. But the direct evidence is still lacking, unless the authors genotype stable transgenic lines (which unfortunately didn't yield such mutations mostly due to low efficiency of Cpf1 in Arabidopsis). So, please tune down this statement.

3. The authors introduced the rationales of tier-1 PCR and tier-2 PCR in discussion (lines 300-304). However, these PCR reactions were used throughout the manuscript. I suggest moving this description earlier when first mentioning these PCR schemes.

4. In the discussion, the authors proposed that TAGs may be more susceptible to such chromosomal rearrangements (e.g., delinver bialleles). However, this may be too speculative, unless the authors also did a control experiment to look at non-TAG genes or random chromosomal regions with multiplexed editing. It may not be TAG specific. This makes me wonder why such delinver bialleles can happen at such high frequencies. Can it be explained by chromosome topology? For example, the homologous pairs of chromosomes may align or physically cluster more than with other random chromosomes. This may happen normally or when DNA repair is activated. In either case, simultaneous deletion of two segments gives each segment two opportunities to invert and ligate, either back to its own chromosome, or to the broken homologous chromosome. As long as the other deletion segment is lost in that cell/plant, one will achieve a delinver biallele. This may explain why such events are of high frequency. Maybe it is worthwhile to investigate this hypothesis using 3C-based techniques. Anyway, I hope the authors can give some insightful discussion along these lines.

5. Finally, it is fair to alarm the readers that we have much to learn about the unexpected chromosomal rearrangements associated with CRISPR mediated multiplexed genome editing. However, the methods used in this study, PCR, has its own limitation. To figure out what really has happened on a genome edited plant, whole genome sequencing will provide a better picture.

Reviewer #3:

Remarks to the Author:

It is well-known that deletions and inversions occur between two CRISPR/Cas targets located at the same chromosome. However, frequencies of inversions and deletion-inversion bi-alleles, demonstrated in this paper, are so high that they are really out of expectation. The finding that frequencies of deletions and inversion mutations are positively correlated with those of deletion mutations is also interesting. This paper will tell readers how frequent of inversion mutations and deletion-inversion bi-alleles are, and thus will provide important guides or references for identification of mutants harboring targeted deletions or inversions of large fragments.

Minor revisions:

Fig. 5c and 5d, it will be better if authors present data involving more combinations of two targets, e.g., gRNA-PE2 and gRNA-PE6.

Response to reviewers' comments

We would like to thank all the reviewers for your critical comments, which helped to greatly improve this work.

Reviewer #1

Reading the manuscript of Liu et al I became more and more surprised by the assessments of the authors. The reason for this is that I find the results by far less surprising as the authors and I also do not agree with a row of their conclusions. I do not agree at all with the conclusion “this is the first report about unintended chromosomal rearrangements induced by CRISPR in plants” (line 317) See e.g.as one case of many the retraction published in Nature Communications <https://doi.org/10.1038/s41467-021-24195-5>, where a large unwanted deletion was found and not a conversion as published before.

Response: We thank the reviewer for correcting our mistake of saying “this is the first report about unintended chromosomal rearrangements induced by CRISPR in plants”. We have deleted such statement and cited the suggested paper and another recent paper (<https://doi.org/10.1002/tpg2.20266>) describing unexpected chromosomal rearrangements induced by CRISPR in plants.

I also think that the more anecdotal findings of the authors that in tandemly arranged genes (TAG) inversions combined with deletions can occur at high frequencies is much less surprising than the author want to make us believe. I guess they were so surprised as they did not take this option into account at all when they first analyzed mutation induction in multiple genes of a tandem array. Obviously, in TAGs such events are more difficult to detect and the paper would well deserve as a note of caution publication in more specialized journal. However, I do see no applicable technical advance nor major new insights in the nature of DSB repair. Indeed, several published reports showed already that not only deletions but also inversions can occur with high frequencies in plants (see e.g. the Schmidt et al., 2019 citation in the manuscript).

Response: We agree with the reviewer that either chromosomal deletions or inversions alone have been reported in plants by earlier studies. However, what we were trying to highlight here is the deletion-inversion (delinver) biallelic mutations, which can escape routine PCR-based genotyping and mislead TAG functional studies. To our knowledge, the delinver biallelic mutations have not been reported so far. As a representative of a large population of plant biologists who are not working in the field of plant chromosomal rearrangements, I was indeed very surprised to see such a high prevalence of delinver mutations in multiplexed CRISPR editing (so were the other two reviewers I guess). In the long list of CRISPR-mediated TAG deletion studies we have cited (Zhou et al., 2014; Durr et al., 2018; Doll et al., 2019; Li

et al., 2019; Shen et al., 2019; Ordon et al., 2020; Xie et al., 2021; Nagy et al., 2021; Li et al., 2022; Géry and Téoulé, 2022), none of these papers have checked the possibility of delinver mutations. In fact, the paper of Shen et al. (2019) is from my own lab, in which we applied the paired gRNA strategy to knock out the TAGs encoding four Arabidopsis metacaspases (AtMCs). We did not realize the existence of such type of mutations until later we stumbled across the delinver issue by chance when knocking out Arabidopsis AtPROPEPs (Figs. 1 and 2). After re-checking the genotypes of *AtMC* quadruple mutant lines, we indeed found that one of the three *AtMC* quadruple mutant lines was actually a delinver bi-allele (Fig. 3c), though the line we have published in the paper of Shen et al. (2019) was luckily an authentic *AtMC* quadruple mutant line (i.e., homozygous deletion line). We suspected that delinver bi-alleles may have been mistakenly used in other TAG deletion studies with the consequence unnoticed. Our work is not meant to bring mechanistic insights into DNA repair or occurrence of delinver mutations. Instead, we aim to raise the alarm about this general phenomenon and help prevent false interpretation of TAG functions due to this hidden genotype issue. Since TAGs constitute a significant part of both plant and human genomes, we believe this work carries important practical relevance and will ignite interest in a broad readership throughout plant research community and will also inspire animal researchers to carefully look into the delinver issue in their TAG functional studies.

The authors speculate that inversion efficiency is especially high within TAGs (line 342) but neither are they able to sustain this hypothesis by comparative experiments nor do they suggest a molecular model to explain this feature.

Response: Thanks for the comment. During revision, we have conducted additional experiments to target two genomic non-TAG regions for deletion in Arabidopsis. One was the adjacent genome region (with a similar size of ~18.2 kb) to the TAG locus of *AT5g45240/AtRPS4/AtRRS1*, which allowed us to compare the delinver frequency of the non-TAG locus with that of its neighboring TAG locus, while the other was a ~32-kb metabolic gene cluster required for triterpene biosynthesis. As a result, we detected comparable levels of delinver mutations at these non-TAG loci as at the TAG loci. These new data (Fig. 6) indicated that delinver mutations are not specific to a TAG locus and are a general phenomenon during genomic deletion. In Discussion, we have tried to provide some speculations on the possible mechanism of how delinver mutations occur. However, as we have explained above, uncovering the mechanism of delinver mutations was not the focus of this study.

The authors also state that they found no correlation between inversion sizes and efficiencies (line 222 following). I am not surprised as the authors have to take the cutting efficiencies of the individual sgRNAs into account, which they don't mention. Indeed, they report on a correlation with simple deletion

efficiencies, which is absolutely in line with cutting efficiencies as decisive factor. Obviously, the more efficient you cut, the higher the chances are to cut not only both sites within a chromosome but also both homologues and obtain besides simple deletions and inversion, deletions in both alleles as well as deletions combined with inversions.

Response: Thanks for bringing up the point about cutting efficiency (gRNA efficiency). Yes, we agree with the reviewer that efficiencies of paired gRNAs determined the deletion frequencies, which in turn were related to the delinver frequencies. Following your suggestion, we now state in the paper “These results hinted that the efficiencies of paired gRNAs, rather than the distance or chromatin features of the target sites, are probably key factors affecting the frequencies of delinver mutations.” Meanwhile, by conducting additional experiments during revision, our new data also suggested that some other parameters in addition to gRNA efficiencies can affect the occurrence of delinver mutations. On one hand, at the rice TAG locus encoding OsPROPEPs, the gRNA-PEP6 in combination with either the gRNA-PEP2 (new data) or gRNA-PEP4 resulted in a total of 32 transgenic calli carrying genomic deletions, but none of them appeared to be delinver bi-alleles (Fig. 5d). On the other hand, at the *Arabidopsis* TAG locus of AT5G45240/*AtRPS4/AtRRS1*, the gRNA-L1/gRNA-R3 pair failed to produce any genomic deletions, whereas the gRNA-L1/gRNA-R4 pair worked efficiently (Fig. 4b), which indicated the ineffectiveness of gRNA-R3. However, our new data (Fig. 6b) showed that, when the gRNA-R3 was paired with the gRNA-L5, 37.5% (39 out of 104) of transgenic plants were found to carry genomic deletions, in which 21 plants (53.8%) represented delinver bi-alleles. These findings underscored unpredictable complexity that successful genomic deletions do not always lead to delinver mutations and that one gRNA can mediate delinver mutations at distinct frequencies when paired with different gRNAs.

Looking at their rice experiments it becomes clear that there is nothing one has to be surprised They find deletions in 49 of 300 cases, which is high but not exceptional: From these 49, 7 contain besides the deletion an inversion. But how many of the pure deletions have both alleles deleted? Most probably much more than inversions. This all absolutely within the ranges reported before for these rearrangements by others and by no means exceptional.

Response: In this work, we did not intend to scare researchers how high the frequency of delinver mutations was in a specific case, as the frequencies of both genomic deletion and delinver mutations can be different for different target loci when different gRNA pairs are used. We only presented the data we obtained by genotyping ~2650 transgenic events, which showed a high prevalence of this hidden genotype issue that can mislead TAG functional studies. Out of 31 pairs of gRNAs that induced large genomic deletions in this study, 27 gave rise to delinver mutations. Although in some cases, the proportion of delinver bi-alleles in deletion-containing plants is moderate,

meaning that there were much more homozygous plants than delinver plants and that the chance of taking the wrong lines for functional study is low. However, in any study without prior knowledge about this hidden genotype issue, the researchers are still facing the risk of taking the wrong lines (delinver bi-alleles) as homozygous deletion alleles for TAG functional assay. We publish this report in order to raise the alarm to the entire plant research community and related human researchers about this hidden genotype issue during multiplexed CRISPR editing. In a real case, no matter the delinver frequency is high or low, we have suggested using a modified three-tier genotyping PCR to identify the bona fide homozygous TAG knockout plants.

Reviewer #2

Due to the easiness of assembly and expression of multiple guide RNAs, multiplexed genome editing by CRISPR systems has been widely used in plant research and applications. However, the potential unexpected consequences of multiplexed editing by CRISPR have not been fully addressed in plants. In this study, Liu et. al made stunning discovery the high frequency deletion-inversion bi-alleles when targeting tandemly arrayed genes (TAGs) for deletion by CRISPR-Cas9 in both Arabidopsis and rice plants. The authors demonstrated their discovery by targeting different TAGs on different chromosomes, and the data is quite convincing. Overall, I found this manuscript very interesting and well written. All the experiments were carefully designed with rigorous analysis. The conclusions are generally sound. I don't have any major criticisms. Rather, I feel the discovery and novelty are quite significant to the field genome editing beyond plants. It is appropriate to consider its publishing in Nature Communications.

Response: We thank the reviewer for the encouragements.

Minor points:

1. The authors mentioned in multiple occasions (e.g., lines 84-85, 307-308, and 316-317) that no studies have investigated the unintended chromosomal rearrangements induced by CRISPR in plants. While I echo the general point by the authors on this surprisingly understudied topic (given the importance of the issue), this claim however is not true anymore. Recently, Zhang et. al reported a genome-wide investigation on off-target effects by multiplexed CRISPR-Cas12a editing in rice (<https://doi.org/10.1002/tpg2.20266>). They discovered unexpected chromosomal rearrangements such as large deletion, duplication, and translocation.

Response: We thank the reviewer for correcting our mistake. We have deleted or rephrased related sentences and have cited the suggested paper and another recent paper (<https://doi.org/10.1038/s41467-022-31001-3>) describing an unexpected chromosomal rearrangement induced by CRISPR in plants.

2. For the Cas12a (Cpf1) editing experiment, since it is done in bulked cells as Arabidopsis protoplasts, it is not possible to tell whether there are, or which are, deletion events. The authors demonstrated inversion can occur with staggered end cutting, which was also demonstrated with ZFNs in Arabidopsis (Qi, et. al G3, 2013). However, their conclusion as stated in lines 332-333 is not accurate because the statement implies that the authors proved deletion events with Cpf1. But the direct evidence is still lacking, unless the authors genotype stable transgenic lines (which unfortunately didn't yield such mutations mostly due to low efficiency of Cpf1 in Arabidopsis). So, please tune down this statement.

Response: Thanks for the suggestion. Indeed, transgenic data are still needed to fully validate the conclusion. We have toned down the statement as suggested.

3. The authors introduced the rationales of tier-1 PCR and tier-2 PCR in discussion (lines 300-304). However, these PCR reactions were used throughout the manuscript. I suggest moving this description earlier when first mentioning these PCR schemes.

Response: Thanks for the suggestion. We now move the rationale of three-tier PCR from Fig. 5e to an earlier place (Fig. 2a), where we have testified the validity of three-tier PCR schemes in distinguishing deletion and deletion mutations and settled down the new PCR schemes for the rest part of the study.

4. In the discussion, the authors proposed that TAGs may be more susceptible to such chromosomal rearrangements (e.g., deletion bialleles). However, this may be too speculative, unless the authors also did a control experiment to look at non-TAG genes or random chromosomal regions with multiplexed editing. It may not be TAG specific. This makes me wonder why such deletion bialleles can happen at such high frequencies. Can it be explained by chromosome topology? For example, the homologous pairs of chromosomes may align or physically cluster more than with other random chromosomes. This may happen normally or when DNA repair is activated. In either case, simultaneous deletion of two segments gives each segment two opportunities to invert and ligate, either back to its own chromosome, or to the broken homologous chromosome. As long as the other deletion segment is lost in that cell/plant, one will achieve a deletion biallele. This may explain why such events are of high frequency. Maybe it is worthwhile to investigate this hypothesis using 3C-based techniques. Anyway, I hope the authors can give some insightful discussion along these lines.

Response: Thank you very much for this comment. As suggested, we have conducted additional experiments to target two genomic non-TAG regions for deletion in Arabidopsis. One was the adjacent genome region (with a similar size: ~18.2 kb) to the TAG locus of *AT5g45240/AtRPS4/AtRRS1*, which

allowed us to compare the delinver frequencies of the non-TAG locus with that of the neighboring TAG locus, while the other was a ~32-kb metabolic gene cluster required for triterpene biosynthesis. As a result, we detected comparable levels of delinver mutations at these non-TAG loci as at TAG loci. These new data (Fig. 6) indicated that delinver mutations are not specific to a TAG locus and are a general phenomenon during genomic deletion. In Discussion, we have tried to provide some speculations on the possible mechanism promoting delinver mutations when genomic deletions take place. From an evolutionary viewpoint, we think that such a mechanism can minimize detrimental impact of genomic deletion on the organismal fitness while increasing genetic variations in progenies to facilitate natural selection-based adaptation (Lee et al., 2017; Wellenreuther and Bernatchez, 2018). Thank you for the great suggestion of using 3C-based techniques to conduct in-depth analysis. However, uncovering the mechanism of delinver mutations was not the focus of this study.

5. Finally, it is fair to alarm the readers that we have much to learn about the unexpected chromosomal rearrangements associated with CRISPR mediated multiplexed genome editing. However, the methods used in this study, PCR, has its own limitation. To figure out what really has happened on a genome edited plant, whole genome sequencing will provide a better picture.

Response: Thanks. As suggested, we have added this good point in Discussion.

Reviewer #3:

It is well-known that deletions and inversions occur between two CRISPR/Cas targets located at the same chromosome. However, frequencies of inversions and deletion-inversion bi-alleles, demonstrated in this paper, are so high that they are really out of expectation. The finding that frequencies of deletions and inversion mutations are positively correlated with those of deletion mutations is also interesting. This paper will tell readers how frequent of inversion mutations and deletion-inversion bi-alleles are, and thus will provide important guides or references for identification of mutants harboring targeted deletions or inversions of large fragments.

Response: We thank the reviewer for the encouragements.

Minor revisions:

Fig. 5c and 5d, it will be better if authors present data involving more combinations of two targets, e.g., gRNA-PE2 and gRNA-PE6.

Response: Thanks for the suggestion. During revision, we have conducted additional experiments to evaluate the frequencies of genomic deletion and delinver mutations between the gRNA-PEP2 and gRNA-PEP6 target sites. As

a result, we observed that 6.6% (20 out of 303) of transgenic rice calli carried genomic deletions between these sites. However, none of these deletion-containing alleles were delivered bi-alleles. The new data are presented in Fig. 5d. As the gRNA-PEP6 in combination with either the gRNA-PEP2 (new data) or gRNA-PEP4 resulted in a total of 32 transgenic calli carrying genomic deletions, but none of them appeared to be delivered bi-alleles (Fig. 5d). These results underscored the complexity that, although delivery mutation frequencies are predominantly associated with efficient genomic deletions, successful genomic deletions do not always lead to delivery mutations. We have added this point in Discussion.

Reviewers' Comments:

Reviewer #1:

Remarks to the Author:

I read the revised version and the rebuttal letter including the other reviews with interest. The authors took up many of the suggestions of the reviewers and also included further data. Thus, the new version of the manuscript is definitely much improved. Indeed – as expected – there is no requirement for a tandem gene setup for the occurrence of delinvers. I still think that that the high correlation between the occurrence of deletions and inversions as alternative outcomes and thus the simultaneous occurrence of both in the two alleles of the same genome is not too surprising for experts but indeed for a wide audience of standard users of GE technology a note of caution that these things happen regularly seems to me justified

Reviewer #2:

Remarks to the Author:

The authors have addressed all my questions in the revision. I have no further comments.

Reviewer #3:

Remarks to the Author:

My concerns were addressed, and I have no further comments

Response to referees

Reviewer #1 (Remarks for the Author)

I read the revised version and the rebuttal letter including the other reviews with interest. The authors took up many of the suggestions of the reviewers and also included further data. Thus, the new version of the manuscript is definitely much improved. Indeed – as expected – there is no requirement for a tandem gene setup for the occurrence of deletions and inversions. I still think that the high correlation between the occurrence of deletions and inversions as alternative outcomes and thus the simultaneous occurrence of both in the two alleles of the same genome is not too surprising for experts but indeed for a wide audience of standard users of GE technology a note of caution that these things happen regularly seems to me justified

Reply: Thanks for agreeing with us that this work is meaningful for a wide audience of common users of genome editing technology.

Reviewer #2:

Comments for the Author:

The authors have addressed all my questions in the revision. I have no further comments.

Reply: Thanks for constructive comments and support.

Reviewer #3:

Comments for the Author:

My concerns were addressed, and I have no further comments.

Reply: Thanks for helpful comments and support.